# Capturing critical *gem*-diol intermediates and hydride transfer for anodic hydrogen production from 5-hydroxymethylfurfural

Guodong Fu[1], Xiaomin Kang[2], Yan Zhang [3], Ying Guo [1], Zhiwei Li[4], Jianwen Liu [1] ✉, Lei Wang[1], Jiujun Zhang [5,6], Xian-Zhu Fu [1] ✉ & Jing-Li Luo[1] ✉

The non-classical anodic $H_2$ production from 5-hydroxymethylfurfural (HMF) is very appealing for energy-saving $H_2$ production with value-added chemical conversion due to the low working potential (~0.1 V vs RHE). However, the reaction mechanism is still not clear due to the lack of direct evidence for the critical intermediates. Herein, the detailed mechanisms are explored in-depth using in situ Raman and Infrared spectroscopy, isotope tracking, and density functional theory calculations. The HMF is observed to form two unique inter-convertible g*em*-diol intermediates in an alkaline medium: 5-(Dihydroxymethyl)furan-2-methanol anion (DHMFM$^-$) and dianion (DHMFM$^{2-}$). The DHMFM$^{2-}$ is easily oxidized to produce $H_2$ via $H^-$ transfer, whereas the DHMFM$^-$ is readily oxidized to produce $H_2O$ via $H^+$ transfer. The increases in potential considerably facilitate the DHMFM$^-$ oxidation rate, shifting the DHMFM$^- \leftrightarrow$ DHMFM$^{2-}$ equilibrium towards DHMFM$^-$ and therefore diminishing anodic $H_2$ production until it terminates. This work captures the critical intermediate DHMFM$^{2-}$ leading to hydrogen production from aldehyde, unraveling a key point for designing higher performing systems.

Due to its high availability, high recyclability, and low cost, the electrocatalytic conversion of organic compounds derived from biomass into value-added products is gaining significance[1]. Oxidizing 5-hydroxymethylfurfural (HMF) can yield valuable chemicals 5-hydroxymethyl-2-furancarboxylic acid (HMFCA), 5-diformylfuran (DFF), 5-formyl-2-furancarboxylic acid (FFCA), and 2,5-furandicarboxylic acid (FDCA) (Fig. 1a)[2], which are very attractive as they may be used to produce a variety of polymers and interleukin inhibitors[3]. Unlike the conventional oxidation approach normally operating under high temperature and high $O_2$ pressure[4], the

electrochemical oxidation of HMF has the advantage of being able to be carried out under mild conditions without the need of an oxidant[5], giving it a wide range of industrial application prospects.

A previous study found that HMF can be electrocatalyzed by a Cu electrode to produce $H_2$ and HMFCA at the anode coupling cathodic hydrogen evolution reaction (HER) with very low potentials (-0.10 V vs. reversible hydrogen electrode (RHE)) to replace the high-potential (>1.23 V) oxygen evolution reaction (OER)[6]. It is very appealing for energy-saving $H_2$ production to be coupled with the value-added chemical conversion[7]. According to a preliminary investigation, the

[1]Shenzhen Key Laboratory of Energy Electrocatalytic Materials, Guangdong Research Center for Interfacial Engineering of Functional Materials, College of Materials Science and Engineering, Shenzhen University, 518060 Shenzhen, China. [2]School of Mechanical Engineering, University of South China, 421001 Hengyang, Hunan Province, China. [3]Pingshan Translational Medicine Center, Shenzhen Bay Laboratory, 518055 Shenzhen, Guangdong Province, China. [4]National Supercomputing Center in Shenzhen, 518055 Shenzhen, China. [5]College of Materials Science and Engineering, Fuzhou University, 350108 Fuzhou, China. [6]Institute for Sustainable Energy, College of Science, Shanghai University, 200444 Shanghai, China. ✉e-mail: jwliu@szu.edu.cn; xz.fu@szu.edu.cn; jll@szu.edu.cn

**Fig. 1 | The conversion of HMF to value-added chemicals and the intermediates. a** Conversion route of HMF to value-added chemicals. **b** The reactions balance of HMF in alkaline medium.

aldehyde group may be responsible for the anodic $H_2$ production. A study of the Cannizzaro reaction in 1887 was the first one to mention the $H_2$ production from the aldehyde group[8]. A few metals or alloys, including Cu[9–13], Au[14], Pd[15,16], and CuAg[9,17], have been discovered to be catalytically or electrocatalytically active for this reaction. Previous mechanistic investigations showed that aldehydes react with hydroxy ions (OH⁻) to generate *gem*-diol intermdiates in an alkaline medium, which was a pre-step for the $H_2$ generation[18]. Isotope tracing experiments showed that the hydrogen atoms of $H_2$ originated from aldehyde group[19–24]. As shown in Fig. 1b, there are three intermediates that should be considered. The aldehyde of HMF can be attacked by $H_2O$, forming the *gem*-diol intermediate 5-(Dihydroxymethyl)furan-2-methanol (DHMFM). The H atom of the hydroxyl of DHMFM can also react with OH⁻, forming the *gem*-diolate ion intermediate (DHMFM⁻). In addition, the HMF can also react with OH⁻, forming DHMFM⁻ directly in an alkanline medium. Moreover, DHMFM⁻ can further react with OH⁻ to generate the *gem*-diolate dianion intermediate (DHMFM²⁻). These reactions are considered the pre-reactions of the Cannizzaro reaction[25]. However, there is still a lack of solid evidence to show how the *gem*-diol intermediates react on the surface of the electrode and produce $H_2$, i.e. the essence of this reaction remains unknown. Furthermore, the electrocatalyst Cu decays extremely fast, preventing its widespread utilization. To promote this reaction in practical industrial applications, a thorough understanding of its reaction mechanism is essential for designing higher performing systems[26,27].

For investigating the possible intermediates on an electrode surface during electrochemical processes. The modern in situ Raman spectroscopy has been extensively used to investigate the mechanism of HER[28], OER[29] and oxygen reduction reaction (ORR)[30]. Moreover, by combining the in situ Raman spectroscopy with density functional theory (DFT) calculations, the structures of the intermediates can be clearly identified so that the mechanisms of the reaction can be fully discovered[31,32].

Herein, the in situ Raman and attenuated total reflectance Infrared (ATR-IR) spectroscopy experiments are carried out to understand the mechanisms of anodic $H_2$ production from HMF. A rough Au-modified Ni foam (Au-Ni) electrode is adopted as an electrocatalyst for the electrochemical HMF oxidation because of the exceptional chemical stability of Au. With the help of DFT calculations and isotope tracing experiments, the critical *gem*-diol intermediates are captured for the

anodic $H_2$ production from HMF on the Au surface. The entire mechanisms are fully discovered. In an alkaline medium, two key inter-convertible *gem*-diol intermediates, i.e., DHMFM⁻ and DHMFM²⁻ are detected on the catalyst surface. DHMFM²⁻ is found to be easily oxidized to produce $H_2$ via H⁻ transfer at low potentials (0.13−0.43 V vs RHE) whereas DHMFM⁻ is found to be readily oxidized to produce $H_2O$ via H⁺ transfer in a wide range of potentials (0.13 − 0.93 V vs RHE). The increases of potentials considerably facilitate the DHMFM⁻ oxidation rate, shifting the DHMFM⁻ ↔ DHMFM²⁻ equilibrium towards DHMFM⁻ and therefore diminishing anodic $H_2$ production until it terminates. As a result, $H_2$ production occurs exclusively at low potentials (0.13−0.43 V vs RHE), whereas $H_2O$ is produced in a wide potential range of 0.13−0.93 V vs RHE. This study provides general mechanistic guidance for further electrocatalytic aldehyde-based $H_2$ production since it represents the widespread occurrence of aldehyde groups in strongly alkaline solutions.

## Results and discussion
### Preparation and characterization of electrocatalyst
The Au−Ni electrodes are prepared by depositing gold nano-cone on Ni foam from an $AuCl_3$ solution. Scanning electron microscopy (SEM), high-resolution transmission electron microscopy (HRTEM), X-ray diffraction (XRD), and X-ray photoelectron spectroscopy (XPS) are used to investigate the morphologies, chemical states, and crystalline of the Au-modified Ni electrode. The SEM images show that Au is evenly deposited on the surface of the Ni foam (Fig. 2a, Supplementary Fig. 1). The XRD shows two distinct patterns (Fig. 2b), in which the peaks at 38.2°, 44.3°, 64.6°, and 77.5° are assigned to Au (PDF# 04-0784), and the peaks at 44.5°, 51.8°, and 76.3° are assigned to Ni (PDF# 04-0850). The high-resolution Au 4*f* XPS spectrum shows two peaks at 83.6 and 87.3 eV (Fig. 2c), which are well fitted to metallic Au[33]. All the characterizations show that the gold nanocones are well deposited on the Ni foam.

Subsequently, the main electrocatalytic activity of the Au-Ni electrode is investigated toward the electrochemical HMF oxidation in the 1 M KOH solution with 0.05 M HMF. As shown in Fig. 2d, the cyclic voltammogram (CV) of the Au-Ni electrode exhibits two oxidation peaks at 0.37 and 0.57 V, indicating the two different species are oxidized. Since the HMF can form DHMFM, DHMFM⁻, and DHMFM²⁻ in an alkaline medium, the peaks should be assigned to the oxidation of

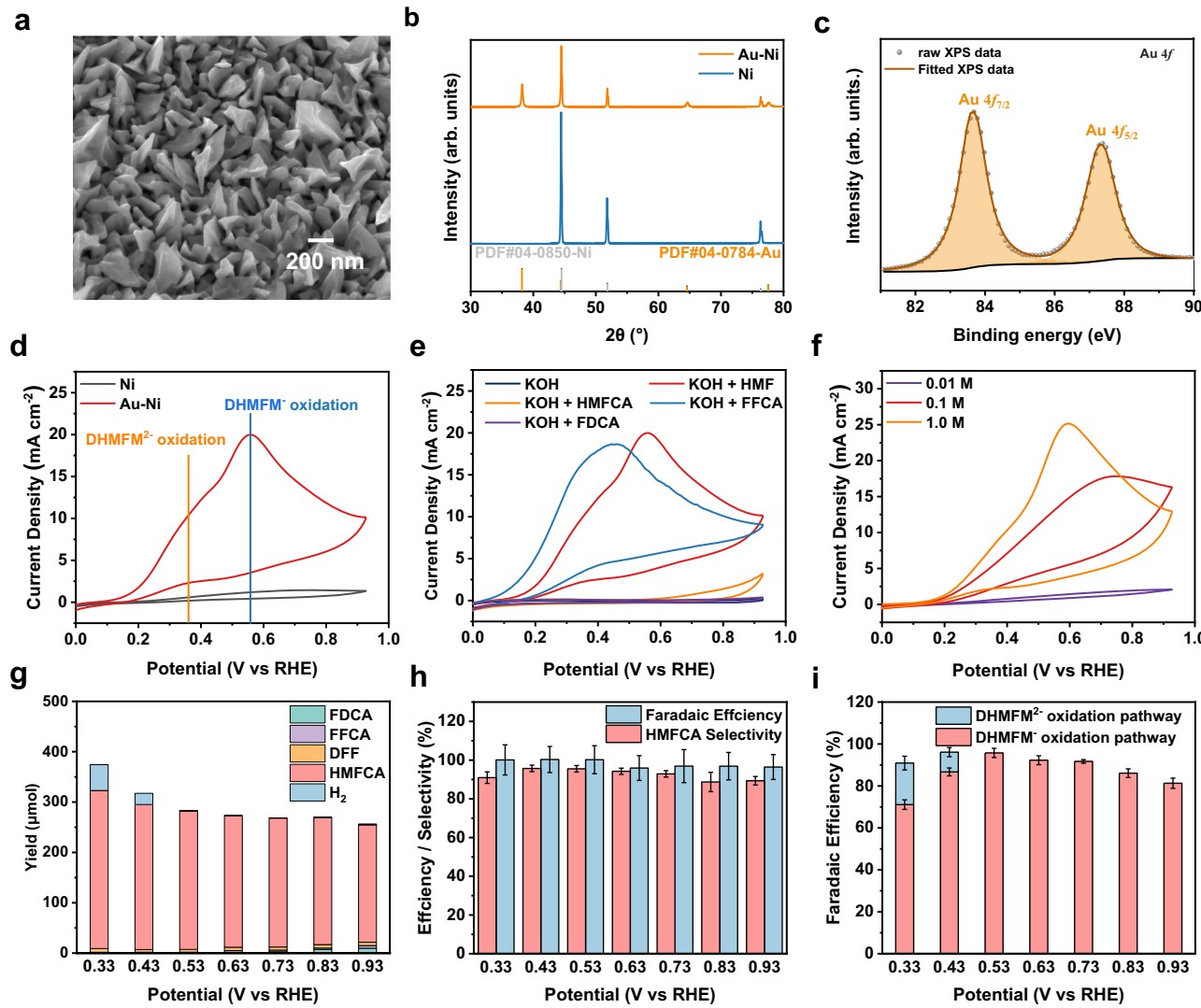

**Fig. 2 | The physical and electrochemical characterization of electrocatalyst.** **a** SEM image of Au–Ni electrode. **b** XRD of Ni foam and Au–Ni electrode. **c** XPS spectra of Au–Ni electrode. **d** CV curve of Au–Ni foam and Ni foam electrode in 1 M KOH with 0.05 M HMF. **e** CV curve of Au-Ni electrode in 1.0 M KOH with and without 0.05 M HMF, HMFCA, FFCA, and FDCA. **f** CV curve of Au-Ni electrode in 0.01 M, 0.1 M, and 1.0 M KOH. **g** Product yield of FDCA, FFCA, DFF, HMFCA, and $H_2$ at various potentials with the same charge. **h** Faradaic efficiency and HMFCA selectivity at various potentials of 0.33−0.93 V. **i** Faradaic efficiency of two pathways at various potentials of 0.33−0.93 V. The potentials reported in this study are referred to RHE unless a special statement is made. The error bars represent the standard deviation for three independent measurements.

these species. However, it is still challenging to identify which intermediate is specifically involved. Notably, the pH value is critical for the oxidation of HMF because the formation of precursors DHMFM⁻ and DHMFM²⁻ usually requires a high concentration of OH⁻, as shown in Fig. 1b. However, in a low alkaline electrolyte, the intermediate DHMFM can be formed since it can be formed by reacting with both $H_2O$ and OH⁻. According to a previous study, the oxidation peak of the electrochemical oxidation reaction of HMF appeared at a higher potential of 1.38 V (vs. RHE) with the product of DFF in a low alkaline electrolyte (pH = 12)[34]. In such an alkaline medium, DHMFM can be clearly formed. However, no oxidation peak was observed below 0.80 V, indicating these peaks cannot be assigned to DHMFM. As a result, the oxidation peaks at 0.37 V and 0.57 V may be attributed to DHMFM²⁻ and DHMFM⁻ oxidation. Since the electrocatalyst Au is deposited on the Ni foams, the Ni foam may influence the result. To rule out the influence, the same CV experiments are performed on bare Ni foam, which displays no current response (Fig. 2d). In addition, a pure Au nanostructure electrode is prepared, and the same CV experiments are performed. The result in Supplementary Fig. 3

exhibits the same electrochemical property as that of the Ni-Au electrode in 1 M KOH and 0.05 M HMF, in which the two oxidation peaks of HMFDM²⁻ and HMFDM⁻ can be clearly distinguished. All these results indicate that Ni foam has no catalytic activity and simply conducts electrons. Furthermore, the same CV experiments are carried out at 1.0 M KOH with 0.05 M furan derivatives HMF, HMFCA, FFCA, and FDCA (Fig. 2e) to determine the oxidation of functional groups involved in the reaction. The FFCA, which has an aldehyde group, exhibits two identical peak responses in a low potential range of 0.10−0.50 V, but the HMFCA, which has a hydroxyl group, seems to have a lower current response in a high potential range of 0.60−1.00 V. The FDCA, which has two carboxyl groups, exhibits no peak responses, comparable to 1.0 M KOH. These results indicate that the aldehyde and hydroxyl groups may be oxidized below 1.00 V whereas the carboxyl group cannot be oxidized below 1.00 V.

The CV test is performed with various concentrations of KOH to analyze how the OH⁻ concentration influences the oxidation behaviors of DHMFM²⁻ and DHMFM⁻. Figure 2f shows that the current response is very low in 0.01 M KOH, a sign of just a small amount of

**Fig. 3 | Two pathways of the HMF oxidation reaction in alkaline medium on Ni–Au electrode.** Two pathways of the HMF oxidation reaction in alkaline medium via (**a**) DHMFM⁻ intermediate oxidation to produce HMFCA and H₂O, and (**b**) DHMFM²⁻ intermediate oxidation to produce HMFCA and H₂.

current flow, implying that only a small amount of the compound is oxidized. As mentioned above, although DHMFM can be easily produced at low OH⁻ concentrations, it will not be oxidized at a voltage below 0.80 V. As a result, it is feasible to conclude that only a small number of intermediates (DHMFM²⁻ and DHMFM⁻) are oxidized. As the OH⁻ concentration increases in the 0.1 M KOH electrolyte, an oxidation peak significantly increases at 0.70 V, indicating that more intermediates are formed and oxidized. When OH⁻ concentration is increased to 1.0 M, two distinct oxidation peaks at 0.37 V and 0.67 V are observed. These two peaks should be assigned to the oxidations of DHMFM²⁻ and DHMFM⁻, respectively. However, further study is still required to confirm the exact assignment of these two peaks.

High-performance liquid chromatography (HPLC) is used to characterize the liquid products, while gas chromatography (GC) is used to characterize the gaseous products. The results (Fig. 2g) show that HMFCA is the major liquid oxidation product, whereas H₂ is the only gas product. The reaction products are sampled at different potentials with the same charge. It is found that the formation of HMFCA increases significantly when H₂ is produced, implying that one of the intermediate oxidations produces more HMFCA with the same charge as compared to the other.

Based on the above experimental observation, two pathways are proposed for electrochemically oxidizing HMF to HMFCA on the Au surface in an alkaline medium, as shown in Fig. 3. HMF initially reacts with OH⁻, producing the *gem*-diol ion intermediates DHMFM⁻ and DHMFM²⁻. The DHMFM⁻ oxidation occurs over a wide potential range of 0.13–0.93 V and produces HMFCA and H₂O while transferring two electrons (Fig. 3a). In comparison, the DHMFM²⁻ oxidation works exclusively at low potential (0.13–0.43 V), yielding HMFCA and H₂ by transferring an electron (Fig. 3b). With this mechanism, the oxidation peaks at 0.37 and 0.57 V in Fig. 2d can be assigned to the oxidation of DHMFM²⁻ and DHMFM⁻, respectively. In addition, the experimental observations in Fig. 2d–g can be well explained using the above-proposed mechanisms.

Furthermore, as the potential increases, the hydroxyl group can be oxidized. It is observed that FFCA is produced at potentials of 0.63–0.93 V, whereas FDCA is produced at higher potentials of 0.73–0.93 V. However, the HMF oxidation reaction has a high HMFCA selectivity of above 85% (Fig. 2h). HMFCA's selectivity decreases very slowly as its potential increases. The main reason is that high voltage can drive both hydroxyl and aldehyde groups to oxidize, resulting in the production of various compounds. As far as the Faradaic efficiency is concerned, it is high at around 100%. Notably, there is a partial Faradaic efficiency even higher than 100% while counting errors (Fig. 2h). The main reason lies in the non-Faradaic process of Cannizzaro reaction, which can also produce HMFCA and H₂ without applying potential. However, through the measurement of the non-Faradaic part of HMFCA (Supplementary Fig. 4), it can be ruled out from the total measured HMFCA. Thus, based on the observation of the

generated hydrogen amount, the corresponding Faradaic efficiencies of the DHMFM⁻ and DHMFM²⁻ oxidation pathways can be determined. Figure 2i shows that the DHMFM⁻ oxidation is the major pathway. The DHMFM²⁻ oxidation pathway reaches a maximum Faradaic efficiency of approximately 20% at a potential of 0.33 V whereas the Faradaic efficiency is near 0 at the potentials of 0.53–0.93 V, indicating that this pathway only takes place in a small port at only low potentials. It confirms the oxidation peak assignment for DHMFM²⁻ and DHMFM⁻ in Fig. 2c–f. Notably, although these experimental observations are consistent with our proposed mechanism, there is still a lack of direct experimental evidence.

### Capturing *gem*-diol intermediates by in situ Raman, ATR-IR spectroscopy and DFT calculations

To obtain the direct evidence for the *gem*-diol intermediates, the electrochemical processes on the gold surface are investigated using in situ Raman spectroscopy and DFT calculations. Since the previous extensive studies by Nørskov et al. showed that the well-crystalline Au surface is often inert while the low-coordinated Au, such as at the sites of edge or cornor, exhibits high activity[35,36], an Au₁₃ cluster is used to simulate the low-coordinated Au surface for modeling the adsorption of HMF and their intermediates on the Au surface.

Supplementary Fig. 5 shows two distinct stages of in situ Raman spectra: the first stage is between 0.03 and 0.43 V, where H₂ is produced, and the second stage is between 0.53 and 0.93 V, where no H₂ is formed. Both DHMFM⁻ and DHMFM²⁻ undergo oxidation during the first stage of the spectra, which falls within the range of 0.13–0.43 V. The in situ Raman spectra for this stage are displayed at the top of Fig. 4a, b. Reactants, intermediates, and products might all coexist in the solution during the electrocatalytic oxidation of HMF. They might also adsorb on the surface of Au. To simulate the adsorption of these species, The HMF (Supplementary Fig. 6), DHMFM²⁻ and DHMFM⁻ (Supplementary Fig. 7), HMFCA (Supplementary Fig. 8), FDCA (Supplementary Fig. 9), and H (Supplementary Fig. 10) are extensively calculated. To fit the experimental data, the well-fitted results for DHMFM²⁻-Au₁₃, DHMFM⁻-Au₁₃, HMFCA-Au₁₃, and H-Au₁₃ are chosen, as indicated in the lower part of Fig. 4a, b. Their summation, i.e., Calc. Sum = DHMFM⁻-Au₁₃ + DHMFM²⁻-Au₁₃ + HMFCA-Au₁₃ + H-Au₁₃, is also compared.

It is found that the summed calculated results (Calc. sum) are in excellent agreement with the experimental results, showing that the Raman experiments detect all the reactants, intermediates, and products. A key takeaway is that the intensity of Au-O peaks at 400–500 cm⁻¹ are greatly enhanced in the in situ Raman spectra. The diverse absorbed species exhibit a diversity of Au–O peak locations, which might be utilized to distinguish among various absorbing species. The most typical Au–O peaks are those of gold oxide (AuO_x) or hydroxy species (Au(OH)_x). However, gold oxide and hydroxy species are usually formed at high voltage such as 0.84 V in 1 M KOH according to the literature[33]. In addition, both of them exhibit Au-O Raman peaks at above 500 cm⁻¹ [37]. Fig. 4a, b makes it very evident that all peaks fall below 500 cm⁻¹, ruling out gold oxides and hydroxides. There are three major peaks below 500 cm⁻¹, i.e., 412 cm⁻¹, 449 cm⁻¹, and 476 cm⁻¹ (Fig. 4a). The peak at 412 cm⁻¹ is attributed to the Au–O stretching of intermediate DHMFM²⁻-Au₁₃ whereas the peak at 449 cm⁻¹ is ascribed to the Au–O stretching of DHMFM⁻-Au₁₃. Another peak at 476 cm⁻¹ is attributed to the Au–O stretching of HMFCA-Au₁₃. As a result, the Au–O peaks indicate that both the intermediates DHMFM⁻, DHMFM²⁻ and product HMFCA are detected, adsorbed on the Au surface.

Figure 4b shows various peaks verifying the adsorption of the intermediates and products for peaks greater than 900 cm⁻¹. The peak at 945 cm⁻¹ belongs to the C–O vibration of DHMFM⁻-Au₁₃, whereas the peak at 1024 cm⁻¹ is ascribed to the C–O vibration of DHMFM²⁻-Au₁₃ and HMFCA. In addition, HMFCA is responsible for the higher peaks at

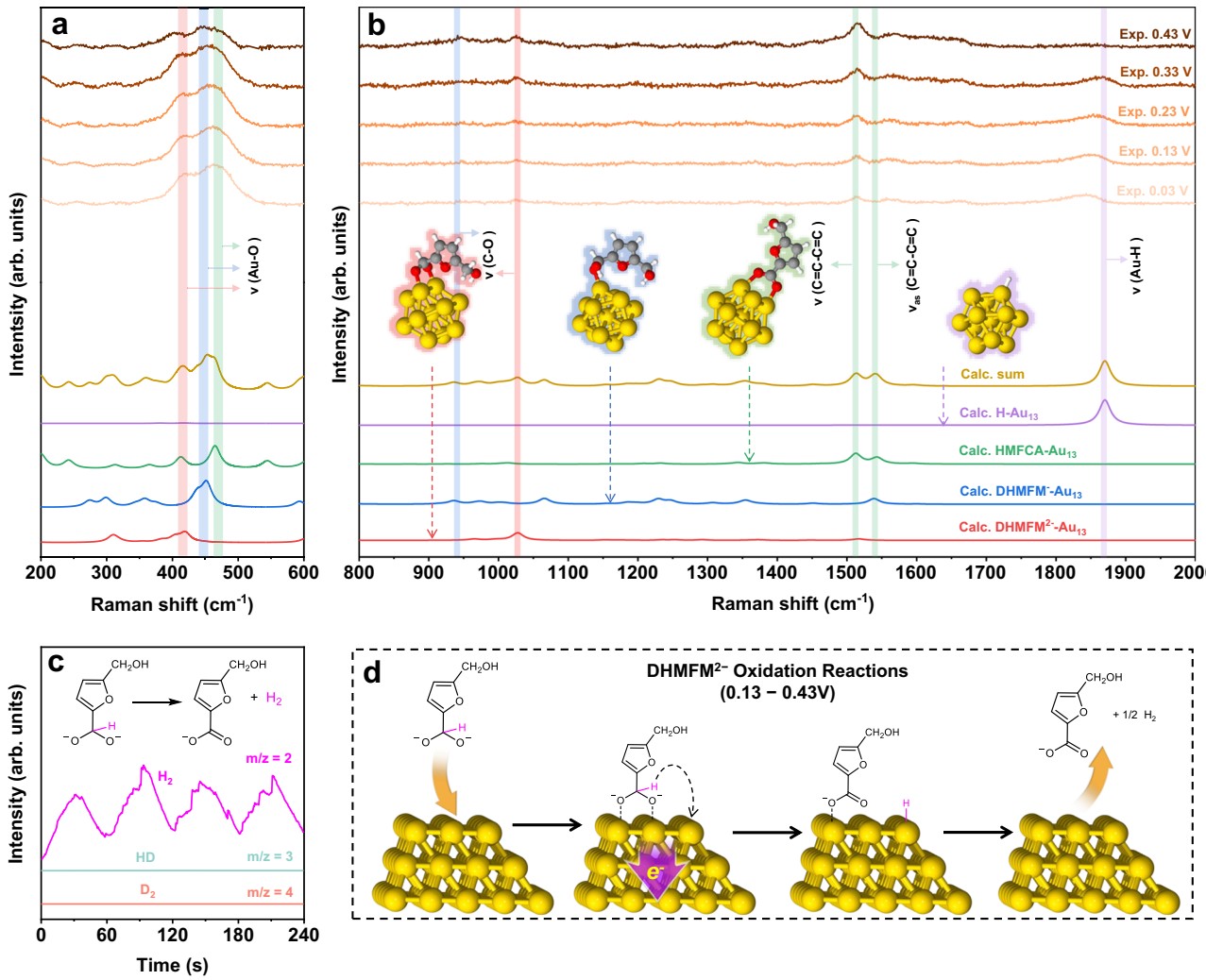

**Fig. 4 | Proposed mechanism of DHMFM²⁻ oxidation reactions according to the experimental and calculated Raman spectra and isotope tracking DEMS Experiment.** In situ Raman spectra 200–600 cm⁻¹ (**a**) and 800–2000 cm⁻¹ (**b**) of HMF oxidation reactions on gold at 0.03–0.43 V (Exp.) and it is fitted by DFT calculated Raman spectra of DHMFM⁻-Au₁₃ (blue), DHMFM²⁻-Au₁₃ (red), HMFCA-Au₁₃ (green) and H-Au₁₃ (purple). **c** DEMS signal of HMF oxidation reaction. **d** Mechanism of the DHMFM²⁻ oxidation pathway on Au–Ni electrode. The color for each element is white for H, gray for C, red for O, yellow for Au.

1512 and 1566 cm⁻¹ according to the vibration mode analysis from DFT calculations. All of these peaks imply the adsorptions of the DHMFM⁻ and DHMFM²⁻ intermediates, as well as the HMFCA product.

Additionally, it is interesting that the Au-H stretching peaks are also observed at 1830–1866 cm⁻¹ depending on the potential. However, it is much lower than that of the HER reaction on the Au-Ni electrode in 1 M KOH (Supplementary Fig. 11). The significant Raman shift change may be attributed to the pH value of the electrolyte and the potential[38]. A high concentration of OH⁻ results in a strong interaction between OH⁻ and the H atoms of Au-H bonds, weakening the Au-H bonds. To verify this, a thorough investigation of the Raman shift based on the Au–H strengthening is conducted using DFT calculation. It shows that the calculated Raman shift of the Au–H bond is inversely correlated with bond length (Supplementary Fig. 10), confirming the experimental observation. The computed results show that the Au–H bond length is slightly adjusted, but the predicted Raman shift is significantly shifted. In alkaline solutions, the interaction between H atoms and OH⁻ is neglected, making direct computation impossible to achieve more accurate results. However, its Raman shift may be approximated by slightly altering the bond length between atoms in the computation, a strategy adopted in this work as well.

In situ ATR-IR spectroscopy is used to confirm the existence of the DHMFM²⁻ and DHMFM⁻ intermediates and the reaction mechanism under the same condition. As shown in Fig. 5a, the ATR-IR spectroscopy contains two types of peaks when compared to the baseline: positive and negative peaks. A positive peak indicates production, whereas a negative number indicates consumption. The peak at 1207 cm⁻¹ is attributed to the C-H rocking of HMFCA (Fig. 5f)[39], which appears from 0.03 V to 0.93 V, indicating HMFCA is produced at this potential. In the range of 0.03–0.63 V, it appears as positive peaks indicating the production of HMFCA, which is due to oxidation from HMFMD²⁻ and HMFMD⁻. However, in the range of 0.73–0.93 V, it appears as negative peaks, indicating the consumption of HMFCA, i.e. HMFCA being further oxidized. Additionally, a positive peak is seen at 1621 cm⁻¹ between 0.43 and 0.93 V. This peak may be attributed to FDCA's C = O stretching[40]. However, it may also be attributed to water since it is close to the bending model of water. It is possible to infer from the information provided by both of those signals that HMFCA is oxidized to produce FDCA. Furthermore, a small negative peak of 1477 cm⁻¹ appears at 0.23 V to 0.43 V, which is attributed to the C-H scissoring of HMFMD²⁻ according to the computational results (Fig. 5b). It indicates the oxidation reaction via HMFMD²⁻. Moreover, a negative peak of 1571 cm⁻¹ is observed from 0.33 V to 0.93 V, which is attributed to the C = C stretching of HMFMD⁻ according to the computational results (Fig. 5c). It indicates an oxidation reaction path via HMFMD⁻. In addition, some negative peaks at 1372, 1522, and 1662 cm⁻¹

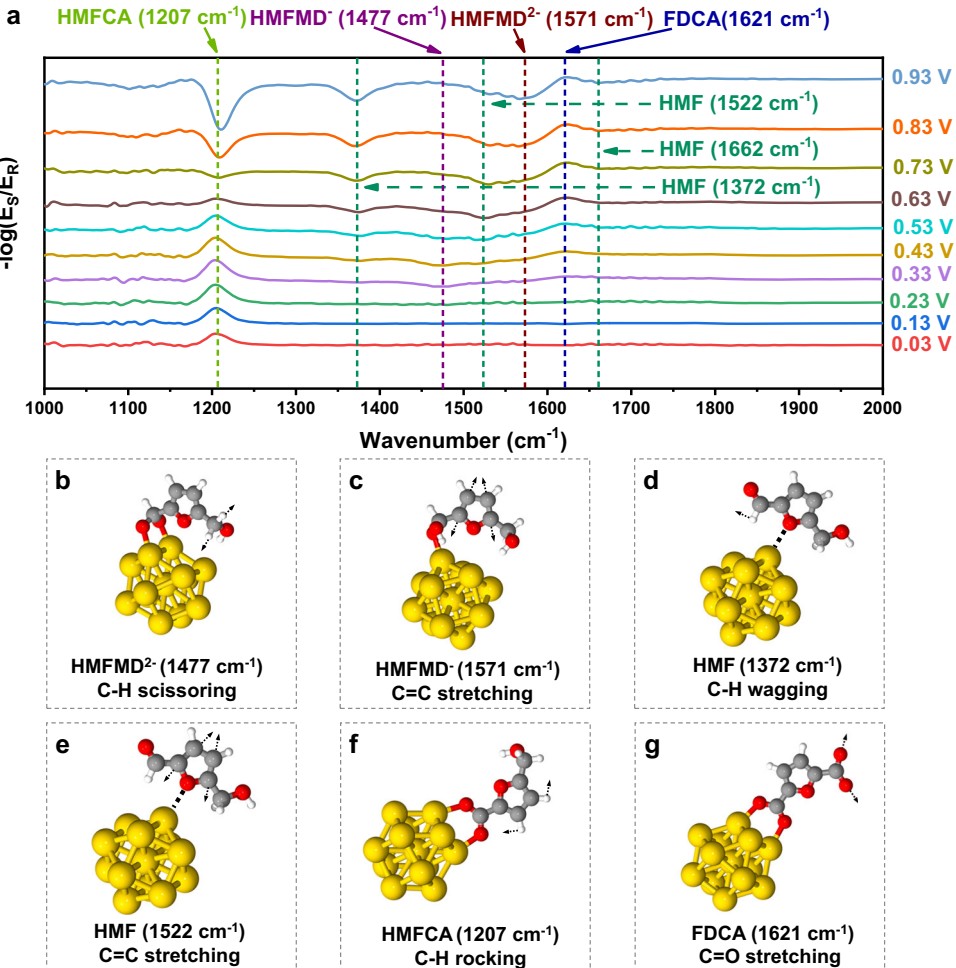

**Fig. 5 | The experimental in situ ATR-IR spectra of the HMF on Ni−Au electrode at 0.03–0.93 V and the vibration mode of the corresponding intermediates.** **a** In situ ATR-IR spectra of HMF oxidation reactions on gold. **b** C−H scissoring of HMFMD²⁻, (**c**) C = C stretching of HMFMD⁻, (**d**) C−H wagging of HMF, (**e**) C = C stretching of HMF, (**f**) C−H rocking of HMFCA and (**g**) C = O stretching of FDCA based on the computational IR frequency analysis. The color for each element is white for H, gray for C, red for O, yellow for Au.

are observed from 0.33 V to 0.93 V, which are attributed to the C-H wagging (Fig. 5d), C = C stretching (Fig. 5e), and C = O stretching (Fig. 5g) of HMF, respectively. According to the aforementioned description, it can be concluded that the results obtained by in situ ATR-IR are in good agreement with the results obtained by in situ Raman.

To further confirm the origins of the produced H₂, an isotope tracing experiment is conducted using deuterated KOD and D₂O. With a chronoamperometry applied to the Au−Ni electrode, an HMF electrochemical oxidation reaction on Au is carried out in 1 M KOD and 0.05 M HMF using D₂O as the solvent (Supplementary Fig. 12). The produced gases are examined using differential electrochemical mass spectrometry (DEMS). Figure 4c illustrates the detected mass spectrometry signals with $m/z$ = 2, 3, and 4, which are attributed to H₂, HD, and D₂, respectively. A working potential of 0.33 V is applied for the DHMFM²⁻ oxdation, but only the signal with $m/z$ = 2 exhibits a quick response, indicating that all the H atoms of H₂ are from DHMFM²⁻ rather than from KOD and D₂O. In comparison, all the signals from $m/z$ = 2, 3, and 4 exhibit responses when a working potential for the HER reaction is applied at −0.37 V (Supplementary Fig. 13), showing all the species involved in the reactions.

The capture of DHMFM²⁻-Au₁₃ and H-Au₁₃ intermediate infers the *gem*-diol intermediate DHMFM²⁻ oxidation mechanism, as shown in Fig. 4d. A DHMFM²⁻ intermediate is initially absorbed on the surface of the gold with two negatively charged O atoms. Similar to the Cannizzaro reaction, the H atom attached to the C atom of *gem*-diol, denoted as the central H atom, is supposed to be negatively charged. The central H atom couples an electron transfer, i.e., the transfer of H⁻ from the CHO₂²⁻ group, which then proceeds in a manner similar to the traditional Cannizzaro reaction. When an H atom is transferred, an electron transfers to the electrode and forms an H−Au bond. The recombination of H atoms produces H₂, which can well explain the anodic H₂ production at low potential.

However, the situation significantly changes at the second stage, with potentials of 0.53–0.93 V. As seen in Fig. 6a,b, the Au-O peak (412 cm⁻¹) and H-Au peak (1830–1866 cm⁻¹) vanish in comparison to the first stage, indicating the disappearance of DHMFM²⁻ and H−Au. Thus, the calculated summation is from DHMFM⁻-Au₁₃ and HMFCA-Au₁₃. A considerable increase in the intensity of the Au−O peak at 449 cm⁻¹ shows that more DHMFM⁻ is being absorbed onto the electrode surface. Additionally, there are other peaks that have been identified as HMFCA-absorbing species, such as C = C (1512 and 1566 cm⁻¹) and C−O (1024 cm⁻¹). It should be noticed that the Raman peak strength of DHMFM⁻ decreases between 0.73 and 0.93 V, with a new Au-O peak appearing at 491 cm⁻¹. This could be attributed to the high potential for the hydroxyl oxidation.

Based on the experimental and calculated results, the oxidation mechanism of DHMFM⁻ is proposed in Fig. 6c, which is quite different from that of DHMFM²⁻. In this mechanism, a DHMFM⁻ intermediate is initially absorbed on the electrode surface, and subsequently, an OH⁻

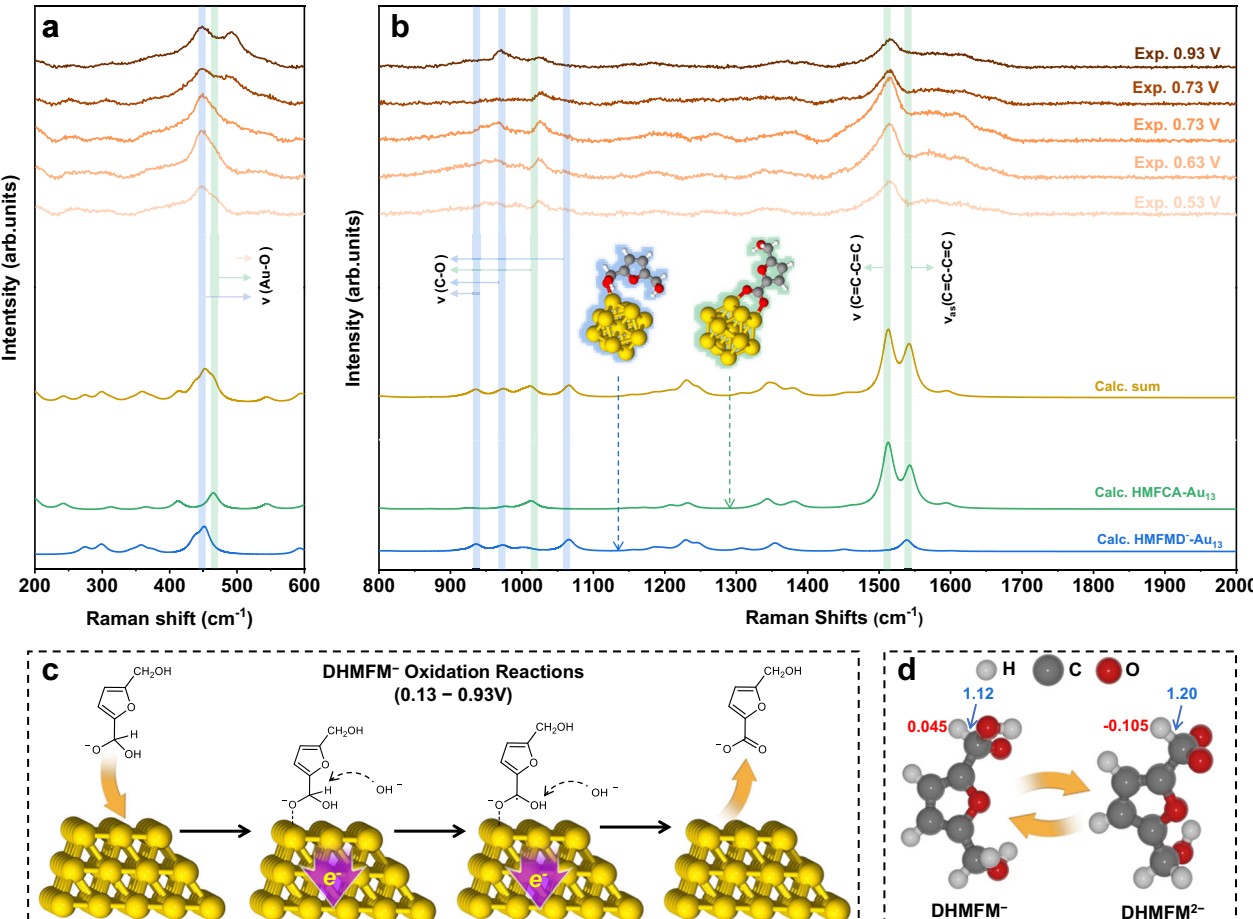

**Fig. 6 | Proposed mechanism of DHMFM⁻ oxidation reactions according to the experimental and calculated Raman spectra.** Experimental in situ Raman spectra 200–600 cm⁻¹ (**a**) and 800–2000 cm⁻¹ (**b**) of HMF oxidation reactions on Au–Ni electrode at 0.53–0.93 V and DFT calculated Raman spectra of DHMFM⁻-Au₁₃ and HMFCA-Au₁₃. **c** Illustration of DHMFM²⁻ oxidation pathway on Au–Ni electrode. **d** Models of optimized DHMFM⁻ and DHMFM²⁻ and calculated charge distribution (red) of the cental-H atoms and length (blue, in Å) of the C–H bonds. The color for each element is white for H, gray for C, red for O, yellow for Au.

ion attacks the central H of the gem-diol intermediate DHMFM⁻, leading to the rupture of the C–H bond. An H⁺ is formed by coupling an electron transfer to the electronode, forming a C(OOH)⁻•. The second OH⁻ then attacks the OH bond of C(OOH)⁻•, forming H₂O and HMFCA via an electron transfer. It is worth noting that the final oxidation products differ due to intermediate differences, i.e., the DHMFM²⁻ intermediate generates HMFCA and H₂ at a low voltage (0.13−0.43 V), whereas the DHMFM⁻ intermediate produces HMFCA and H₂O throughout a wide voltage range (0.13−0.93 V).

To uncover the intrinsic difference between the mechanisms of DHMFM⁻ and DHMFM²⁻, the charge distributions are calculated as shown in Fig. 6d. The central H atom of the *gem*-diol DHMFM⁻ intermediate is positively charged, whereas the central H atom of the *gem*-diol DHMFM²⁻ is negatively charged. Due to the central H atom's changing charge, the associated C–H bond length varies. The C–H bond length for DHMFM⁻ is 1.12 Å, but for DHMFM²⁻, it is weakened to 1.20 Å. The difference in the charge of the central hydrogen atom and the bond length of the corresponding C–H bond leads to their significantly different transfer behaviors in electrochemical reactions. Positively charged H may be transferred as an H atom, i.e., an H⁺ is transferred along with an electron transfer, and H⁺ easily interacts with OH⁻ to produce H₂O due to electrostatic attraction. In contrast, the negatively charged H might be transferred as H⁻, i.e., an H atom is coupled with an electron transfer. The H atom is transferred on the Au surface to produce an Au−H bond, which then becomes H₂. As a result, the two intermediates undergo completely different reaction

processes. Understanding the underlying cause of the discrepancy is essential to uncovering the entire mechanism.

The fundamental difference between the two reactions originates from their two intermediates, and there is an equilibrium between these two intermediates, i.e. DHMFM⁻ ↔ DHMFM²⁻. The reason may be ascribed to the equilibria of DHMFM²⁻ and DHMFM⁻ (Fig. 2b). According to the reaction constant of a typical electrocatalytic process[41], the reaction rate is positively correlated to the applied potentials, i.e., the higher the applied potential, the higher the reaction rate. In this system, neither the solvent nor the electrocatalyst changes. The only variable that has changed is the applied potential. As the potential increases, so does the rate of DHMFM⁻ conversion to HMFCA and H₂O, and that of DHMFM²⁻ conversion to HMFCA and H₂. However, the formation of DHMFM²⁻ is from the reaction between DHMFM⁻ and OH⁻. So the fast consumption of DHMFM⁻ results in a low concentration of DHMFM²⁻ at the electrode surface (Fig. 7).

As a consequence of this process, DHMFM²⁻ near the electrocatalyst is greatly decreased, resulting in lower H₂ production at the anode. This process illustrates that anodic H₂ generation is mostly determined by the DHMFM⁻ ↔ DHMFM²⁻ equilibrium, as illustrated in Fig. 7. In a strong alkaline solution, the aldehyde group usually reacts with OH⁻, therefore, the equilibrium between the *gem*-diol anion and the *gem*-diol dianion may always exist independent of the electrocatalyst. As a result, the aldehyde and electrocatalyst used are the two factors that may influence anodic H₂ production. Since the accumulation of electrons on the central hydrogen atom can promote the

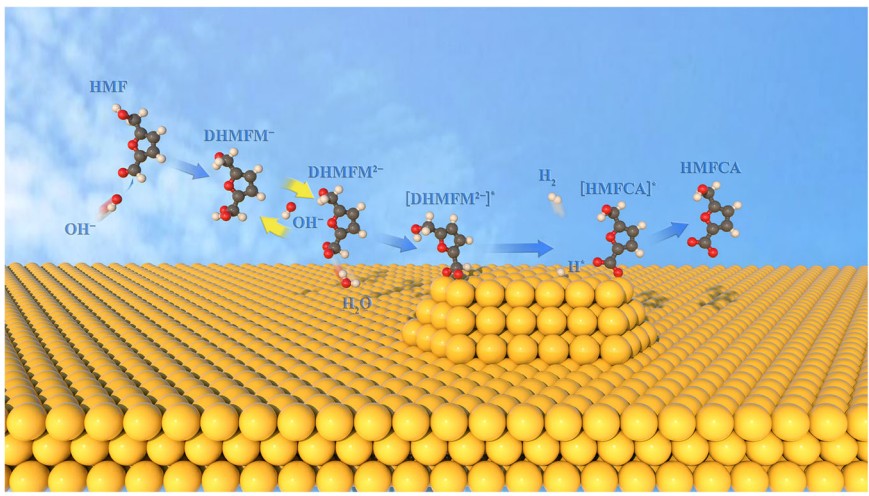

**Fig. 7 | Illustration of the reaction on the suface of Ni-Au electrode.** Illustration of the DHMFM⁻ ↔ DHMFM²⁻ equilibrium and anodic H₂ production. The color for each element is white for H, gray for C, red for O, yellow for Au.

reaction to produce H₂ via H⁻ transfer, the precise construction of aldehydes is a good strategy to promote H₂ production. On the premise of avoiding other side reactions, the electron-donating group can be chosen to bond to the formaldehyde group, which can promote electron accumulation on the central H atom of the *gem*-diol intermediate, thereby promoting H₂ production. Furthermore, since there is an equilibrium between the *gem*-diol ainon and the *gem*-diol dianion, a high-selectivity electrocatalyst may be designed to inhibit the reaction of the central H atom in the single-anioned *gem*-diol via H atom transfer to facilitate H⁻ transfer of the *gem*-diol dianion into H₂. From the above in-depth analysis, it can be clearly seen that the proposed mechanism in this work provides a general mechanistic guidance for electrocatalytic aldehyde-based hydrogen production. To confirm the validity of this mechanism, Cu is employed as the electrocatalyst and in situ Raman spectroscopy is used to analyze the reaction. The results further support our proposed mechanisms. Supplementary Fig. 17, 18 illustrate the comprehensive results, while the detailed analysis is provided in the supplementary information.

The detailed mechanism of anodic hydrogen production from the oxidation of HMF on the gold surface is established by capturing two unique inter-convertable *gem*-diol intermediates, single-anioned DHMFM⁻ and double-anioned DHMFM²⁻ using in situ Raman and ATR-IR spectroscopy, isotope tracking, and DFT calculations. The DHMFM⁻ is discovered to produce H₂O via H⁺ coupling electron transfer ascribing to the positively charged central H atom. In comparison, the DHMFM²⁻ is ready to produce H₂ via a hydrogen coupling electron transfer process, i.e., H⁻ transfer at the anode due to the negatively charged central H atom of DHMFM²⁻. At low voltages, since both DHMFM⁻ and DHMFM²⁻ depleting reactions are relatively slow, neither reaction completely overwhelms the other, allowing both reactions to be observed. However, at high voltages, the conversion of single-anioned DHMFM⁻ is greatly increased, shifting the DHMFM⁻ ↔ DHMFM²⁻ equilibrium towards DHMFM⁻ and inhibiting the formation of DHMFM²⁻ and subsequent anodic H₂ production. As a result, the anodic H₂ production electrocatalyzed by Au is only observed at low potentials. According to the mechanism study, the main strategies that may be implemented to increase the hydrogen generation of an electrocatalytic aldehyde-based anode primarily involve two factors: The first factor is the electron-donating group, which could be employed to bond to the aldehyde group, enhancing the electron accumulation on the central H atom to form a negatively charged central H atom for the *gem*-diol intermediate. The second one is the development of high-selectivity electrocatalysts, which suppress H⁺ transfer but favor H⁻ transfer. This study provides general

mechanistic guidance for electrocatalytic aldehyde-based hydrogen production.

## Methods
### DFT computational details
Adsorption on Au₁₃ gold cluster is used to mimic the absorbed Raman spectroscopies of HMF, intermediates DHMFM⁻, DHMFM²⁻ and H, and products HMFCA and FDCA. Geometric optimizations and frequency calculations are carried out using the B3LYP hybrid functional[42,43] and def2-TZVP basis sets[44,45], as implemented in Gaussian09 Revision D01[46]. The partial optimization for charged molecules adsorbed on Au₁₃ cluster is conducted with the fixation of Au₁₃ cluster, relaxing the others to avoid the reconstruction of Au₁₃ induced by charge transfer from charged molecules. All of the structures' frequencies are analyzed, and all of the frequencies are real, showing that all of the structures are local minimums. To match the experiment and earlier investigations, computational spectra are scaled by a factor of 0.978.

### Preparation of the Au nano cone array electrode
The Au nanocone array electrode is prepared according to the literature[47]. Commercial Ni foam (1 × 1 cm) is cleaned for 10 min in 1 M HCl, ethanol, and deionized (DI) water. After cleaning with DI water, the Ni foam is dried at 60 °C for further use. 1 ml of a 50 mM HAuCl₄ solution is mixed with 4 ml of 25 mg/ml Cetyltrimethylammonium Bromide (CTAB) solution, causing a rapid color change from colorless to opaque orange-red. A piece of Ni foam is immersed in the solution for 12 h under room temperature (approximately 25 °C). Then, the Au nanocone array electrode is obtained. The electrode is taken out and washed with ethanol and DI water, respectively. The electrode is then dried at 40 °C for further use. The mass loading of the Au on the electrode is measured as 2.4 mg cm⁻² by Inductively Coupled Plasma-Optical Emission Spectrometer.

### Preparation of the pure Au nano-structured electrode
Before the electrodeposition, the gold foil is washed with ethanol and DI water three times and dried at 40 °C. A 10 × 10 × 0.1 mm gold foil is used as the working electrode, a Pt foil as the counter electrode, and Ag/AgCl as the reference electrode. Gold nanostructures are electrodeposited onto gold foil at −0.4 V vs. Ag/AgCl for 300 seconds in a 50 mM HAuCl₄ solution under room temperature (approximately 25 °C). Then, the electrode is washed by DI water three times and dried in air for further use. The mass loading of the gold is the change of the mass of the electrode before and after the electrodeposition. It is 1.4 mg cm⁻².

## Preparation of the pure Cu nano-structured electrode

A piece of copper foam is cut to a size of $1 \times 1$ cm$^2$ and washed with ethanol and DI water for 5 min, respectively. Then, the copper foam is immersed in a static 2 M NaOH/ 0.11 M ammonium persulphate aqueous solution for 10 min under room temperature (approximately 25 °C) to form a Cu(OH)$_2$ nanoneedle on the surface of the copper foam. After that, the copper foam is immersed in 0.1 M NaBH$_4$ for 10 min under room temperature (approximately 25 °C) to reduce Cu(OH)$_2$ into Cu. Finally, the copper foam is taken out, washed with DI water for 15 min, and dried at room temperature for further use.

## General characterization

The morphologies of electrodes are characterized by field-emission scanning electron microscopy (FESEM, SU-70) and field-emission transmission electron microscopy (FETEM, JEM-F200). TEM analysis is conducted using a JEOL JEM-F200 field emission microscope equipped with an EMSIS Xarosa CCD camera. X-ray diffraction (XRD) patterns are recorded using the Bruker D8 Advance (Cu Kα, 50 kV and 360 mA). X-ray photoelectron spectroscopy (XPS) is conducted on a Thermo Scientific™ K-Alpha™+ spectrometer equipped with a monochromatic Al Kα X-ray source (1486.6 eV) operating at 100 W. Samples are analyzed under vacuum ($P < 10^{-8}$ mbar) with a pass energy of 150 eV (survey scans) or 25 eV (high-resolution scans). All the peaks are calibrated with C1s peak binding energy of 284.8 eV for adventitious carbon.

## Electrochemical experiments

All electrochemical measurements are conducted with an electrochemical workstation (CHI760E, CH Instruments Inc., Shanghai) at room temperature (approximately 25 °C). The electrochemical measurements are carried out in a three-electrode system. The catalyst-loaded Ni foam ($1 \times 1$ cm$^2$) is used as the working electrode, and Hg/HgO is used as the reference electrode. A Pt sheet ($1 \times 1$ cm$^2$) is used as a counter electrode. 10 ml 1 M KOH with or without 0.05 M HMF or other intermediates is used as an electrolyte, and polarization curves are collected with a scan rate of 10 mV/s. All potentials presented in this work are calibrated to the reversible hydrogen electrode (RHE) according to the Eq. (1):

$$E_{RHE} = E_{Hg/HgO} + 0.0591 \times pH + 0.098 \tag{1}$$

The double-layer capacitance ($C_{dl}$) of the as-prepared electrode is measured in 1 M KOH by cyclic voltammetry in a potential range of 0.127–0.227 V vs RHE with a scan rate of 10–50 mV/s. IR-compensation was not performed throughout the electrochemical measurement.

## Product analysis

The identification and quantification of HMF and other liquid oxidation products are performed by high-performance liquid chromatography (HPLC) on Shimadzu Prominence-II LC 2030 Plus. After a certain number of charges had been passed, 30 μL of the electrolyte solution is withdrawn and diluted with 1470 uL DI water. The HPLC is equipped with a Shim-pack GIST column (5 μm C18, 4.6 ×250 mm) operated at a fixed temperature of 25 °C and a UV-VIS detector set at λ = 265 nm. 70% 5 mM ammonium formate aqueous solution and 30% methanol are used as eluents. An injection volume of 10 μL is applied, and the identification and concentration of the yield of liquid oxidation products are calculated based on the following Eqs.(2) and (3):

$$Yield = concentration \ of \ product \times volume \ of \ electrolyte \ (10 \ ml) \tag{2}$$

In this electrochemical system, there are two Faradaic processes, i.e., HMFMD$^{2-}$ and HMFMD$^-$ oxidation pathways. However, some non-Faradaic processes, such as the Cannizzaro reaction, can also produce HMFCA. Considering non-Faradaic processes, an experiment without applied potential is conducted to measure and rule out this part of HMFCA. The ratio of the HMFCA generated by the non-Faradaic process to the Faradaic process ranges from 7.9% (0.33 V) to 0.4% (0.93 V) depending on potentials.

$$Yield \ of \ HMFCA = total \ HMFCA - HMFCA \ of \ non \ Faraday \ process \tag{3}$$

The Faradaic efficiency of product formation and HMFCA selectivity are calculated based on the following Eqs. (4) and (5):

$$EF(\%) = \frac{Yield \ of \ product \times n \times F}{total \ charge \ passed} \times 100\% \tag{4}$$

Where n is the number of electrons transferred for each product formation and F is the Faradaic Constant (96485 C/mol).

$$HMFCA \ selectivity \ (\%) = \frac{Yield \ of \ HMFCA}{Yield \ of \ all \ products} \times 100\% \tag{5}$$

The concentration of detected products during HMF oxidation reactions at different potentials is shown in Supplementary Fig. 14. The standard curve of HMF and products is shown in Supplementary Fig. 15.

Gas chromatography (GC) is used to identify and quantify gas products using the Fuli GC2010 Plus, which is outfitted with a thermal conductivity detector and used Ar as the carrier gas. Supplementary Fig. 14 shows the standard GC of H$_2$ gas.

The Faradaic efficiency of the DHMFM$^{2-}$ oxidation pathway and the DHMFM$^-$ oxidation pathway according to the amount of product H$_2$ and HMFCA may be calculated quantitatively using the Eqs. (6), (7) and (8):

$$FE_{(HMFMD^{2-})} = \frac{2n_{(H_2)}F}{Q} \times 100\% \tag{6}$$

$$FE_{(HMFMD^-)} = \frac{2n_{(O-HMFCA)}F}{Q} \times 100\% \tag{7}$$

$$n_{(O-HMFCA)} = n_{(HMFCA)} - n_{(H-HMFCA)} = n_{(HMFCA)} - 2n_{(H_2)} \tag{8}$$

Where $FE_{(HMFMD^{2-})}$, $FE_{(HMFMD^-)}$ are the Faradaic efficiency for the DHMFM$^{2-}$ and DHMFM$^-$ oxidation pathways, respectively. $n_{(H2)}$, $n_{(O-HMFCA)}$, $n_{(H-HMFCA)}$ and $n_{(HMFCA)}$ are the moles of produced hydrogen, HMFCA produced by DHMFM$^-$ oxidation pathway, HMFCA produced by DHMFM$^{2-}$ oxidation pathway, and HMFCA produced in total, respectively. The $F$ and $Q$ are Faradaic constant (96465 C mol$^{-1}$) and the quantity of electric charge, respectively.

## DEMS experiment

An in situ DEMS set-up (Shanghai Linglu Instrument Equipment) is employed for the measurement, with a Teflon film separating the electrolyte from the vacuum system to minimize aqueous solvents entering the mass spectrometer. The vacuum system consists of two dry pumps and one turbo pump, and the vacuum is maintained below $2 \times 10^{-4}$ Pa. The preparation process for the working electrode is the same as for the Au nanocone array electrode. The Au nanocone array electrode ($1 \times 1$ cm$^2$) is used as the working electrode, and Hg/HgO is used as the reference electrode. A Pt sheet ($1 \times 1$ cm$^2$) is used as a counter electrode. 10 ml 1 M KOH with 0.05 M HMF is used as an electrolyte. The experiment is carried out under room temperature (approximately 25 °C)

## In situ Raman spectroscopy experiment

Raman spectroscopy is performed with a Renishaw Invia Raman system under room temperature (approximately 25 °C). A 785 nm laser is utilized for all experiments. A custom-built one-compartment poly (ether-ether-ketone) electrochemical cell (Gaossunion (Tianjin) Photoelectric Technology Co.) with a transparent quartz window is used for in situ electrochemical Raman spectroscopy experiments. A Pt wire counter electrode and a custom Hg/HgO reference electrode are employed. An Au nanocone electrode ($1 \times 1$ cm²) or Cu nano structured electrode ($1 \times 1$ cm²) is used as a working electrode, and 50 ml 1 M KOH with 0.05 M HMF solution is used as an electrolyte. Raman spectra are collected either in open circuit or during a chronoamperometric run after the system had been allowed to stabilize for 5 min at each potential.

## In situ ATR-IR experiment

A Thermo Nicolet 8700 spectrometer equipped with an MCT detector cooled by liquid nitrogen is employed for the electrochemical ATR-IR (Otto) measurements. The measurements are carried out under room temperature (approximately 25 °C). A Si prism is used as the internal reflection element. The Au nano cone array electrode ($1 \times 1$ cm²) is used as the working electrode, Hg/HgO as the reference, which is introduced near the working electrode via a Luggin capillary, and a Pt mesh (1 cm × 1 cm) serves as the counter electrode. 50 ml 1 M KOH with 0.05 M HMF is used as the electrolyte, and the distance between the working electrode and the Si prism is about 1 μm. All spectra are shown in $-\log(\frac{E_s}{E_R})$, with $E_s$ and $E_R$ representing the sample and reference spectra, respectively. The reference spectra are corrected in an open circuit condition, and the sample spectra are collected at different potentials. The spectral resolution is 4 cm⁻¹ for all the measurements, unless otherwise mentioned.

## Data availability

Figure 2, Figs. 4–6 data generated in this study are provided in the Source Data file. Source data are provided with this paper.

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

## Acknowledgements

The authors gratefully thank the financial support from the National Natural Science Foundation of China (22272108, J.W.L; 21975163, X.Z.F and 22003041, X.M.K.), Shenzhen Science and Technology Program (KQTD20190929173914967, J.L.L.; ZDSYS20220527171401003, J.L.L.) and the Senior Talent Research Start-up Fund of Shenzhen University (000263 X.Z.F. and 000265J.W.L.). We sincerely acknowledge the Instrumental Analysis Center of Shenzhen University (Xili Campus) for HRTEM measurements. G.D. Fu thanks Dr. Xiaohui Deng for useful discussions.

## Author contributions

G.D.F. performed the experiments, contributed to the DFT calculations, and wrote the manuscript. X.M.K. and Y.Z. contributed to the TEM and Raman analysis. Z.W.L. contributed to the DFT calculations. G.Y., L.W., and J.J.Z. proposed some key mechanisms and revised the manuscript. J.W.L., X.Z.F., and J.L.L. conceived the idea, supervised the project, and edited the manuscript.

## Competing interests

The authors declare no competing interests.
