## [Peer Review File · Nature Communications]

REVIEWER COMMENTS

Reviewer #1 (Remarks to the Author):

Fu et al. captured two inter-convertible gem-diol intermediates using in situ Raman spectroscopy, isotope tracking and DFT calculations. The experimental procedure is reasonable and the results could convey a conclusive message for the mechanistic insight. However, some conclusions are too subjective, and some scientific problems require further revision.

1. SEM and TEM images are too blurred.
2. Figure 1h, counting the error, why is there a partial FE higher than 100%?
3. How do the authors distinguish the FE of HMFMD2- and HMFMD- oxidation pathways? Is it by hydrogen production rate? The vibration peaks of DHMFM2--Au13 (1024 cm⁻¹) are also present in Figure 3a-b, so the oxidation of HMFMD2- still exists at the potential of 0.53-0.93 V. Therefore, the absence of H₂ production at 0.53-0.93 V may be attributed to the higher potential and cannot be fully considered as the effect of intermediates.
4. In page 9, line 203, the oxidation peaks are shown in Figure 1d, not Figure 1c. Furthermore, there are many mistakes in detail; please check the whole manuscript carefully.
5. It is recommended to add in situ IR to make the results more convincing.

Reviewer #2 (Remarks to the Author):

In this manuscript, the authors studied the mechanism of 5-Hydroxymethylfurfural electrooxidation on gold for anodic hydrogen generation with Raman, GC, HPLC, DFT as well as DEMS. A lot of work has been done. However, there are some flaws in the manuscript as shown below. I would suggest reconsidering to publish the manuscript after a major revision.

1. For the DEMS measurements, I did not find the experiment's details in the manuscript.
2. Lines 297-303, typos, the mass charge ratio should be m/z or m/e instead of m/s.
3. Figures 2c, S9 and S10a, the authors claimed that 0.33 V vs 0.17 V (vs. RHE) was used. However, in both Figures S9 and S10, the applied potentials were the same between 0.37 and 0.13 V, but the mass spectra signals were quite different. At positive potentials, there should be positive currents

for oxidation reactions. However, the currents showed in Figures were negative. Did you purge solutions with N₂ or Ar?

4. Lines 301-303, at 0.17 V (vs. RHE), the hydrogen evolution reaction on gold cannot happen at this positive potential.

5. Again, in Figure S1, the currents were negative in both scan directions. You might have had oxygen in the solution.

6. It is well known that aldehyde electrooxidation on 1b metals such as Au, Ag and Cu in alkaline media generates hydrogen, and has been studied for many decades. There are many relevant papers that may be helpful for mechanism study, however, were not cited by the authors.

Reviewer #3 (Remarks to the Author):

In the submitted manuscript, the authors attempted to decipher the anodic hydrogen production mechanism during HMF electro-oxidation to HMFCA on gold in basic solutions. In such conditions, HMF, DHMFM, DHMFM⁻, and DHMFM₂⁻ are present in the bulk state. Previous studies suggested that the anodic H₂ production probably originated from the aldehyde group. In this study, anodic H₂ production is directly linked to the DHMFM₂⁻, and specifically the Au-H bond formation at low overpotentials. Also, the DHMFM⁻ ↔ DHMFM₂⁻ equilibrium dictates the extent of anodic H₂ production.

The proposed mechanism is reasonable but I would not consider the level of experimental and theoretical support fully sufficient as this is the main driving point of the paper. Several suggestions to improve and questions are below.

Facet dependence – how does the surface facet influence the Au-O modes and positions in the Raman spectra. Would this also cause different Au-O vibrations as well as different intermediates?

Could the Raman spectra and hypothesis be verified with metals like Au, Ag, and Pd which have recently been shown to be close to 100% selective for the anodic pathway that produces H₂? I would assume then, if the mechanism proposed in the paper is correct, that the DHMFM₂⁻ would be the main intermediate detected in the Raman spectrum. Luckily these materials are also SERS active. This would also help to determine if the proposed insights are generalizable.

It seems Figure S11 is mistakenly referred as S12 in line 477. Please double check all the figures.

How thick is the Au catalyst on the Ni. Even if Ni is not active for this, are there any interfacial effects or synergy in a way that Ni augments the activity of Au? This could be verified by depositing the Au on a surface like carbon paper and testing performance.

For Fig. S11 – H₂ production should also be measured.

In the abstract it is mentioned that “The main challenge lies in the rapid electrocatalysts’ performance decay, which necessitates a comprehensive understanding of the reaction mechanism to develop novel electrocatalysts.” Please mention that how this proposed mechanism can be used to address the electrocatalyst stability issue. Is this different than previous systems that were reasonably stable for the anodic H₂ production/aldehyde production pathway?

Response Letter to Referees

Manuscript ID: NCOMMS-23-05443

Dear editor and referees,

We sincerely thank all the reviewers and editor for the valuable comments and suggestions. In this Response Letter to Referees, we have addressed and clarified all the comments/concerns raised by the reviewers. With substantial amount of work having been added into the revised manuscript according to the reviewers' comments, the quality of our manuscript has been significantly improved and we hope that the revised manuscript can now meet the high standards of *Nature Communications* and be published in your prestigious journal.

Listed below are our point-by-point responses to the reviewers' comments, and all the related revision are highlighted in the revised manuscript.

REVIEWER COMMENTS

Reviewer #1 (Remarks to the Author):

Fu et al. captured two inter-convertible *gem*-diol intermediates using in situ Raman spectroscopy, isotope tracking and DFT calculations. The experimental procedure is reasonable and the results could convey a conclusive message for the mechanistic insight. However, some conclusions are too subjective, and some scientific problems require further revision.

Reply: Thanks for the comments. Following the suggestions, we have made extensive revisions to correct the scientific problems, and conducted a series of experiments, such as *in situ* IR and Raman, which more convincingly prove the proposed mechanisms and further confirm our conclusion.

1. SEM and TEM images are too blurred.

Reply: Thanks for the comment. To solve the blurred image problem, a high-resolution SEM image is used to show the main morphology. The other TEM images that show the detailed morphology are moved to **Figure S1** in the Supplementary Information (SI). The revised **Figure 1** and **Figure S1** are shown as follows:

Figure 1. (a) SEM image of Au-Ni electrode. (b) XRD of Ni foam and Au-Ni electrode. (c) XPS spectra of Au-Ni electrode (d) CV curve of Au-Ni foam and Ni foam electrode in 1 M KOH with 0.5 M HMF. (e) CV curve of Au-Ni electrode in 1.0 M KOH with and without 0.5 M HMF, HMFCA, FFCA and FDCA. (f) CV curve of Au-Ni electrode in 0.01M, 0.1M and 1.0M (g) Product yield of FDCA, FFCA, DFF, HMFCAs and H₂ at various potentials with the same charge. (h) Faradic efficiency and HMFCAs selectivity at various potentials of 0.33–0.93V. (i) Faradic efficiency of two pathways at various potentials of 0.33–0.93V. The potentials reported in this study are referred to RHE unless specified otherwise.

Figure S1 (a-b) SEM images of an Au-Ni electrode (c) TEM image of the Au-Ni electrode surface, and (d) Fourier transform image of the selected area (white square area) in the TEM image.

2. Figure 1h, counting the error, why is there a partial FE higher than 100%?

Reply: Thanks for the comment. The partial FE of higher than 100% is due to a measurement error in the quantity of HMFCA produced through non-Faradaic processes. In this electrochemical system, there are two Faradaic processes, i.e., HMFMD^{2-} and HMFMD^- oxidation pathways. However, there exist some non-Faradaic processes, such as the Cannizzaro reaction that can also produce HMFCA, which makes partial FE higher than 100%. To clarify this, the following paragraph is added to the manuscript (Page 9, line 213-218):

As far as the Faradaic efficiency is concerned, it is high at around 100%. Notably, there is a partial Faradaic efficiency even higher than 100% while counting errors (**Figure 1h**). The main reason lies in the non-Faradaic process of the Cannizzaro reaction, which can also produce HMFCA and H_2 without applying potential. However, through the measurement of the non-Faradaic part of HMFCA (**Figure S4**), it can be ruled out from

the total measured HMFCA.

In addition, the following revision is made in the section of **Product analysis** (Page 23, line 525-529):

In this electrochemical system, there are two Faradaic processes, i.e., HMFMD²⁻ and HMFMD⁻ oxidation pathways. However, some non-Faradaic processes, such as the Cannizzaro reaction, can also produce HMFCA. Considering non-Faradaic processes, an experiment without applied potential is conducted to measure and rule out this part of HMFCA.

3. How do the authors distinguish the FE of HMFMD²⁻ and HMFMD⁻ oxidation pathways? Is it by hydrogen production rate?

Reply: Thanks for the comment. These two reactions are distinguished by the amount of H₂ produced. To clarify it, the following revision is added to the manuscript (Page 10, line 218-221):

Thus, based on the observation of the generated hydrogen amount, the corresponding Faradaic efficiencies of the DHMFM⁻ and DHMFM²⁻ oxidation pathways can be determined. **Figure 1i** shows that the DHMFM⁻ oxidation is the major pathway.

4. The vibration peaks of DHMFM²⁻-Au₁₃ (1024 cm⁻¹) are also present in Figure 3a-b, so the oxidation of HMFMD²⁻ still exists at the potential of 0.53-0.93 V. Therefore, the absence of H₂ production at 0.53-0.93 V may be attributed to the higher potential and cannot be fully considered as the effect of intermediates.

Reply: Thanks for the comment. We checked the attribution of the vibration peak in 1024 cm⁻¹ with our calculation result and experimental result in the literature (*J. Raman Spectrosc.* 2011, **42**, 2069–2076). The peak of 1024 cm⁻¹ can be attributed to the C-O stretching vibration of the hydroxy group (-CH₂OH) ¹ not only for DHMFM²⁻ but also for HMFCA. Its existence at the potential of 0.53-0.93 V is mainly attributed to HMFCA because the difference between DHMFM²⁻ and HMFCA can be clearly distinguished by the characteristic Au-O vibration peaks. As a result, the absence of H₂

production at 0.53-0.93 V should be attributed to the effect of intermediates. To clarify it, the assignment of 1024 cm^{-1} is revised in Table S1, and the following revision is made in the manuscript (Page 12, line 278-280):

The peak at 945 cm^{-1} belongs to the C-O vibration of DHMFM⁻-Au₁₃, whereas the peak at 1024 cm^{-1} is ascribed to the C-O vibration of DHMFM²⁻-Au₁₃ and HMFCA.

5. In page 9, line 203, the oxidation peaks are shown in Figure 1d, not Figure 1c. Furthermore, there are many mistakes in detail; please check the whole manuscript carefully.

Reply: Thanks for the comment. We have corrected this error by changing “Figure 1c” to “Figure 1d”. In addition, we also checked the entire manuscript again and changed “m/s” to “m/z”

6. It is recommended to add in situ IR to make the results more convincing.

Reply: Thanks for the good suggestion. We conducted an *in-situ* attenuated total reflectance Infrared (ATR-IR) experiment with the Ni-Au electrode. The results (Figure 3) show that HMFMD²⁻ and HMFMD⁻ intermediates can also be observed. And the electrochemical behaviors of these two intermediates also correspond with the electrochemical experiment and *in-situ* Raman experiment in our manuscript. To clarify it, the following contents have been added to the manuscript (Page 13, line 298-326):

In situ ATR-IR spectroscopy is used to confirm the existences of the DHMFM²⁻ and DHMFM⁻ intermediates and the reaction mechanism under the same condition. As shown in Figure 3a, the ATR-IR spectroscopy contains two types of peaks when compared to the baseline: positive and negative peaks. A positive peak indicates production, whereas a negative number indicates consumption. The peak at 1207 cm^{-1} is attributed to the C-H rocking of HMFCA,^[34] which appears from 0.03 V to 0.93 V, indicating HMFCA is produced at this potential. In the range of 0.03–0.63 V, it appears as positive peaks indicating the production of HMFCA, which is due to oxidation from HMFMD²⁻ and HMFMD⁻. However, in the range of 0.73–0.93 V, it appears as

negative peaks, indicating the consumption of HMFCA, i.e. HMFCA being further oxidized. Additionally, a positive peak is seen at 1621 cm^{-1} between 0.43 and 0.93 V, which is attributable to FDCA's C=O stretching.^[35] It is possible to infer from the information provided by both of those signals that HMFCA is oxidized to produce FDCA. Furthermore, a small negative peak of 1477 cm^{-1} appears at 0.23 V to 0.43 V, which is attributed to the C-H scissoring of HMFMD²⁻ according to the computational results (**Figure 3b**). It indicates the oxidation reaction via HMFMD²⁻. Moreover, a negative peak of 1571 cm^{-1} is observed from 0.33 V to 0.93 V, which is attributed to the C=C stretching of HMFMD⁻ according to the computational results (**Figure 3c**). It indicates an oxidation reaction path via HMFMD⁻. In addition, some negative peaks at 1372 , 1522 , and 1662 cm^{-1} are observed from 0.33 V to 0.93 V, which are attributed to the C-H wagging, C=C stretching, and C=O stretching of HMF, respectively. According to the aforementioned description, it can be concluded that the results obtained by *in situ* ATR-IR are in good agreement with the results obtained by *in situ* Raman.

In addition, the method for *in situ* ATR-IR experiment is added to manuscript in the “***In situ* ATR-IR experiment**” part as follows (Page 24, line 572-582):

A Thermo Nicolet 8700 spectrometer equipped with an MCT detector cooled by liquid nitrogen is employed for the electrochemical ATR-IR (Otto) measurements. A Si prism is used as the internal reflection element. The Ni-Au electrode is used as the working electrode, Hg/HgO as the reference, which is introduced near the working electrode via a Luggin capillary, and a Pt mesh ($1\text{ cm} \times 1\text{ cm}$) serves as the counter electrode. 1 M KOH with 0.05 M HMF is used as the electrolyte, and the distance between the working electrode and the Si prism is about $1\text{ }\mu\text{m}$. All spectra are shown in $\frac{\Delta R}{R} = \frac{E_S - E_R}{E_R}$, with E_S and E_R representing the sample and reference spectra, respectively. The spectral resolution is 4 cm^{-1} for all the measurements, unless otherwise mentioned.

Figure 3. (a) In-situ ATR-IR spectra of HMF oxidation reactions on gold. (b) C-H scissoring of HMFMD²⁻, (c) C=C scissoring of HMFMD⁻, (d) C-H wagging of HMF, (e) C=C stretching of HMF, (f) C-H rocking of HMFCFA and (g) C=O stretching of FDCA based on the computational IR frequency analysis.

Reviewer #2 (Remarks to the Author):

In this manuscript, the authors studied the mechanism of 5-Hydroxymethylfurfural electrooxidation on gold for anodic hydrogen generation with Raman, GC, HPLC, DFT as well as DEMS. A lot of work has been done. However, there are some flaws in the

manuscript as shown below. I would suggest reconsidering to publish the manuscript after a major revision.

Reply: Thanks for the valuable comments and suggestions. Based on the suggestions, we have carefully and extensively revised the manuscript, and corrected the flaws. In addition, a series of experiments, such as *in situ* IR and Raman, are conducted and the results convincingly prove the proposed mechanisms and further confirm the conclusion.

7. For the DEMS measurements, I did not find the experiment's details in the manuscript.

Reply: Thanks for the comments. We added experimental details of DEMS measurement in the methods sections as follows (Page 24, line 555-561):

An in situ DEMS set-up (Shanghai Linglu Instrument Equipment) is employed for the measurement, with a Teflon film separating the electrolyte from the vacuum system to minimize aqueous solvents entering the mass spectrometer. The vacuum system consists of two dry pumps and one turbo pump, and the vacuum is maintained below 2×10^{-4} Pa. The preparation process for the working electrode is the same as for the Au nanocone array electrode.

8. Lines 297-303, typos, the mass charge ratio should be m/z or m/e instead of m/s.

Reply: Thanks for the comment. We are sorry for the typo. We have carefully checked our manuscript again and changed “m/s” to “m/z”.

9. Figures 2c, S9 and S10a, the authors claimed that 0.33 V vs 0.17 V (vs. RHE) was used. However, in both Figures S9 and S10, the applied potentials were the same between 0.37 and 0.13 V, but the mass spectra signals were quite different. At positive potentials, there should be positive currents for oxidation reactions. However, the currents showed in Figures were negative. Did you purge solutions with N₂ or Ar?

Reply: We are sorry for these mistakes. We double checked the original data and found

that we applied a different potential between 0.13 V and -0.37 V, this is a potential range where the HER is able to occur. The Figure S13 (i.e. old Figure S10) is corrected as below:

Figure S13 (a) Potential signal and (b) Differential Electrochemical Mass Spectrometry (DEMS) signal of hydrogen evolution reaction on Au electrode in 1 M KOD and 0.5 M HMF.

10. Lines 301-303, at 0.17 V (vs. RHE), the hydrogen evolution reaction on gold cannot happen at this positive potential.

Reply: Thank you for commenting. We sincerely apologize for this mistake. As previously stated, the correct potential values for the HER reaction should be -0.37 V (vs. RHE). The manuscript has been revised to correct the mistake.

11. Again, in Figure S1, the currents were negative in both scan directions. You might have had oxygen in the solution.

Reply: Thank you for commenting. We repeated the CV test by purging the electrolyte with Ar to remove the oxygen in the solution, and the result indicates no oxygen in the solution this time. **Figure S2** (i.e. old Figure S1) illustrates the new results.

Figure S2 (a) CV curve of Ni-Au electrode in 1 M KOH. (b) Linear fitting of the C_{dl} for Au electrode.

12. It is well known that aldehyde electrooxidation on Ib metals such as Au, Ag and Cu in alkaline media generates hydrogen, and has been studied for many decades. There are many relevant papers that may be helpful for mechanism study, however, were not cited by the authors.

Reply: Thanks for the comments and suggestion. Following the suggestion, we made a revision as follows (Page 4, line 71-73): A few metals or alloys, including Cu^[9-13], Au^[14], Pd^[15-16], and CuAg^[9,17], have been discovered to be catalytically or electrocatalytically active for this reaction. In the revised manuscript, the recent relevant papers are cited as follows:

[10] Pan, Y. et al. Unveiling the synergistic effect of multi-valence Cu species to promote formaldehyde oxidation for anodic hydrogen production. *Chem* 2023, 9, 963.

[11] Yang, Y. et al. A direct formaldehyde fuel cell for CO₂-emission free Co-generation of electrical energy and valuable chemical/hydrogen. *Angew. Chem. Int. Ed.* 2023, 135, e202302950.

[17] Li, G. et al. Dual hydrogen production from electrocatalytic water reduction coupled with formaldehyde oxidation via a copper-silver electrocatalyst. *Nat. Commun.* 2023, 14, 525.

Reviewer #3 (Remarks to the Author):

In the submitted manuscript, the authors attempted to decipher the anodic hydrogen production mechanism during HMF electro-oxidation to HMFCFA on gold in basic solutions. In such conditions, HMF, DHMFM, DHMFM⁻, and DHMFM²⁻ are present in the bulk state. Previous studies suggested that the anodic H₂ production probably originated from the aldehyde group. In this study, anodic H₂ production is directly linked to the DHMFM²⁻, and specifically the Au-H bond formation at low overpotentials. Also, the DHMFM⁻ ↔ DHMFM²⁻ equilibrium dictates the extent of anodic H₂ production. The proposed mechanism is reasonable but I would not consider the level of experimental and theoretical support fully sufficient as this is the main driving point of the paper. Several suggestions to improve and questions are below.

Reply: Thanks for the comments and suggestions. Some additional results of *in situ* IR and Raman are included to support the proposed mechanisms. In addition, the proposed mechanism is validated by a Cu-catalyzed reaction. All of these additional results reinforce the conclusion.

13. Facet dependence – how does the surface facet influence the Au-O modes and positions in the Raman spectra. Would this also cause different Au-O vibrations as well as different intermediates?

Reply: Thanks for the comment. This is a significant topic. It is frequently crucial for a catalytic reaction to have facet selectivity. We take this issue seriously for this system as well. Different crystal planes, however, were shown to have minimal impact in practical trials. We discovered, after extensive literature study, that Nørskov et al. investigated the activity of the crystal face of Au in detail. The low-coordinated Au, i.e. the Au in the edges or the corners, is the one that is active rather than the ideal Au crystal face, which is inactive. In this article, our simulation modeling is likewise built on this basis. In order to imitate low-coordination Au, we constructed a model of Au₁₃. To clarify it, the following revision is made in the revised manuscript (Page 10, line 232-

236):

Since the previous extensive studies by Nørskov et al. show that the well-crystalline Au surface is often inert while the low-coordinated Au, such as edge sites or corner sites exhibit high activity, an Au₁₃ cluster is used to simulate the low-coordinated Au surface for modeling the adsorption of HMF and their intermediates on the Au surface.

14: Could the Raman spectra and hypothesis be verified with metals like Au, Ag, and Pd which have recently been shown to be close to 100% selective for the anodic pathway that produces H₂? I would assume then, if the mechanism proposed in the paper is correct, that the DHMFM²⁻ would be the main intermediate detected in the Raman spectrum. Luckily these materials are also SERS active. This would also help to determine if the proposed insights are generalizable.

Reply: Thanks for the comments and suggestion. To verify the proposed mechanism, a Cu nano-structure electrode is synthesized and an in-situ Raman spectroscopy experiment is conducted since copper is the best catalyst for aldehyde hydrogen evolution reactions with approximate 100% selectivity and it is also reported as the surface enhancing Raman spectra material. It was found that The Raman spectra and proposed mechanisms can indeed be verified by Cu. The following contents have been added to the manuscript (Page 19, line 426-430):

To confirm the validity of this mechanism, Cu is employed as the electrocatalyst and *in situ* Raman spectroscopy is used to analyze the reaction. The results further support our proposed mechanisms. **Figures S17 and S18** illustrate the comprehensive results, while the detailed analysis is provided in the **Supplementary information**.

In addition, the synthesis method for the Cu nano-structure is added in the manuscript as follows (Page 21, line 482-489):

Preparation of the pure Cu nano structured electrode

A piece of copper foam is cut to a size of 1 x 1 cm² and washed with ethanol and DI water for 5 minutes, respectively. Then, the copper foam is immersed in a static 2 M NaOH/ 0.11 M APS aqueous solution for chemical oxidation to form a Cu(OH)₂

nanoneedle on the surface of the copper foam. After that, the copper foam is immersed in 1 M NaBH₄ for 10 minutes to reduce Cu(OH)₂ into Cu. Finally, the copper foam is taken out, washed with DI water for 15 minutes, and dried at room temperature for further use.

The following contents have been added in the SI (Page 20-22 of SI):

As shown in **Figure S17** a and b, the surface of the Cu foam is covered by Cu nanoneedles with a diameter of 200 nm. Such a nanostructure with a large surface area is beneficial to both electrocatalysis activity and the surface-enhancing Raman effect. **Figure S17c** shows the XRD pattern of the Cu electrode, in which the peaks at 43.3°, 50.4°, and 74.0° are assigned to Cu (PDF#04-0836). **Figure 17d** shows the electrochemical behavior of the Cu nanostructure electrode in 1M KOH and 0.05 M HMF. It clearly shows that the HMF hydrogen production reaction occurs at 0–0.5 V, whereas the partial Cu is oxidized to Cu₂O at 0.5–0.6 V. However, both Cu and Cu₂O are oxidized to Cu(OH)₂ at a potential above 0.6 V.

An in-situ Raman experiment is conducted on the Cu nano-structure electrode, and the results are shown in **Figure S18**, which is different from the spectra conducted on the gold nano-structure electrode. The main difference lies in the signals for the intermediates HMFDM²⁻ and HMFDM⁻. In the in-situ Raman experiment for the Cu nano-structure electrode, the peak at 400 cm⁻¹ is attributed to the Cu-O of HMFDM²⁻ absorbed species. The other typical Cu-O peaks, which have been widely studied, can be assigned to Cu₂O (150, 220, 415, 520, 630 cm⁻¹), CuO (303, 350, 636 cm⁻¹), Cu(OH)₂ (292, 488 cm⁻¹) and CuO₂⁻ (636 cm⁻¹). (*ACS Catal.* 2016, **6**, 2473–2481) However, there is no Cu-O peak for HMFDM⁻ observed, indicating that no HMFDM⁻ pathway occurs on the surface of the Cu electrode. The observation is in line with the performance results. In addition, The Cu-H peak, which is considered the most important intermediate for the production of H₂, is also observed at 2066 cm⁻¹. When the potential rises to 0.33V-0.43V, some new peaks appear, and the new peak at 558 cm⁻¹ is attributed to the Cu-O of HMFCA absorbed species. Notably, the intensities of HMFDM²⁻ and HMFCA peaks decrease since the Cu electrode is oxidized to Cu₂O above 0.53 V and

Cu₂O is not active in hydrogen production from aldehyde. Meanwhile, the peaks of Cu-H at 2066 cm⁻¹ disappear at this point, confirming the stoppage of hydrogen production from aldehyde. As a result, DHMFM²⁻ is the main intermediate of the Cu electrode-catalyzed aldehyde hydrogen production reaction, which not only explains why the selectivity of H₂ is close to 100% on Cu but also verifies the proposed mechanisms.

Figure S17. SEM images of as synthesis Cu nanostructure electrode in Low (a) and High (b) magnification. (c) The XRD pattern of Cu nanostructure electrode. (d) The LSV curve of Cu nanostructure electrode in 1 M KOH and 0.05 M HMF.

Figure S18. *In situ* Raman spectrum of Cu nanostructure electrode in 1 M KOH and 0.05 M HMF

15. It seems Figure S11 is mistakenly referred as S12 in line 477. Please double check all the figures.

Reply: Thanks for the comments. The error was corrected and the all the Figures were double checked again.

16. How thick is the Au catalyst on the Ni. Even if Ni is not active for this, are there any interfacial effects or synergy in a way that Ni augments the activity of Au? This could be verified by depositing the Au on a surface like carbon paper and testing performance.

Reply: Thanks for the comment and suggestion. As suggested by the reviewer, we designed an additional experiment to test the effect of Ni. A pure gold electrode was prepared by electrodepositing gold nano structures on a gold foil. The electrode was then characterized and employed to conduct an electrochemical experiment. the results show that Ni has no obvious effect on the electrocatalytic effect. To clarify it, the

following revision has been made (Page 6, line 148-153):

In addition, a pure Au nanostructure electrode is prepared, and the same CV experiments are performed. The result shown in **Figure S3** exhibits the same electrochemical property as that of the Ni-Au electrode in 1M KOH and 0.05 M HMF, in which the two oxidation peaks of HMFDM^{2-} and HMFDM^- can be clearly distinguished. All these results indicate that Ni foam has no catalytic activity and simply conducts electrons.

Figure S3. SEM images of as synthesis Au nanostructure electrode in low (a) and high (b) magnification. (c) The XRD pattern of Cu nanostructure electrode. (d) The LSV curve of Cu nanostructure electrode in 1 M KOH and 0.05 M HMF.

In addition, the prepared method is added as following (Page 21, line 475-481):

Preparation of the pure Au nano structured electrode

Before the electrodeposition, the gold foil is washed by ethanol and DI water three times and dried at 40 °C. A 10 x 10 x 0.1 mm gold foil is used as the working electrode,

a Pt foil as the counter electrode, and Ag/AgCl as the reference electrode. Gold nanostructures are electrodeposited onto gold foil at -0.4V vs. Ag/AgCl for 300 seconds in a 50 mM H AuCl_4 solution. Then, the electrode is washed by DI water three times and dried in air for further use.

17. Fig. S11 – H_2 production should also be measured.

Reply: Thanks for the suggestion. We measured the H_2 production on Ni-Au electrode without applying potential by GC and the results are listed in **Figure S4** (i.e. the old Figure S11) . However, GC does not show any H_2 signals as shown in **Figure S4**.

Figure S4. (a) GC curve from 1 M KOH and 0.05 M HMF without applied potential. (b) Yield of HMFCa and H_2 in 1 M KOH and 0.05 M HMF without applied potential.

18. In the abstract it is mentioned that “The main challenge lies in the rapid electrocatalysts’ performance decay, which necessitates a comprehensive understanding of the reaction mechanism to develop novel electrocatalysts.” Please mention that how this proposed mechanism can be used to address the electrocatalyst stability issue. Is this different than previous systems that were reasonably stable for the anodic H_2 production/aldehyde production pathway?

Reply: Thanks for the comment. The critical intermediate DHMF^{2-} for hydrogen production from aldehyde is captured in this work. Only by catalyzing the critical

intermediate DHMFM²⁻ stably can hydrogen be produced. To clarify it, the following revision is made in abstract part: This work captures the critical intermediate DHMFM²⁻ leading to hydrogen production from aldehyde, unraveling a key point for electrocatalyst design to remedy the performance decay.

REVIEWER COMMENTS

Reviewer #1 (Remarks to the Author):

The revised manuscript is acceptable for publication.

Reviewer #2 (Remarks to the Author):

I would recommend this manuscript for publication after minor revision.

1. Please check the potential values in Figures S12 and S13.
2. Line 337, it should be -0.33 V instead of -0.37 V. Is the solvent D2O?
3. In Figure S12 and line 333, the potential should be 0.37 V instead of 0.33 V. Is the solvent D2O?
4. Line 330, it should be Figure S12 instead of Figure S13.
5. As I remember, Baltruschat, Jusys and Heitbaum studied the hydrogen evolution mechanism during formaldehyde and acetaldehyde electrooxidation at Ib metals (Cu, Ag and Au) by using DEMS several decades ago. I suggest the authors cite them. The following is an incomplete list: Electrochim. Acta 2002, 47, 4485-4500 (10.1016/S0013-4686(02)00521-2); Electrochim. Acta 1993, 38, 1067-1072 ([https://doi.org/10.1016/0013-4686\(93\)80214-K](https://doi.org/10.1016/0013-4686(93)80214-K))
6. In Figure 3, how were the reference spectrum/spectra taken? Please check the Y axis. Is it absorbance or reflectance? The relevant text seemed to mean that the Y axis is absorbance, however, the reflectance was shown in Figure 3. The band at 1621 cm⁻¹ could also be assigned to the HOH bending mode of water.

Reviewer #3 (Remarks to the Author):

The authors adequately addressed all concerns. Minor comments left below:

The authors discuss performance decay and how the elucidation of reaction intermediates can help solve this issue. I think this is overstated as the performance decay is often an intrinsic property of the catalyst itself rather than of specific intermediates, except in the cases of surface poisoning. Probably better to edit the arguments in the abstract/intro – I would think capturing the mechanism is more important towards modelling, designing higher performing systems and so on.

From figure S4 – the amount of HMFCAs produced from non-Faradaic solution reactions is quite high (3 mM in 2 hrs). How does this quantity compare to the HMFCAs produced through Faradaic processes? This should be stated.

Typos – please fix throughout text (e.g. Faradic vs Faradaic)

Response Letter to Referees

Manuscript ID: NCOMMS-23-05443A

Dear editor and referees,

We sincerely thank all the reviewers and editor for the valuable comments and suggestions. In this Response Letter to Referees, we have addressed and clarified all the comments/concerns raised by the reviewers. With a careful revision according to the reviewers' comments, the quality of our manuscript has been significantly improved and we hope that the revised manuscript can now meet the high standards of *Nature Communications* and be published in your prestigious journal.

Listed below are our point-by-point responses to the reviewers' comments, and all the related revision are highlighted in the revised manuscript.

REVIEWER COMMENTS

Reviewer #2 (Remarks to the Author):

I would recommend this manuscript for publication after minor revision.

Reply: Thanks for the valuable comment.

1. Please check the potential values in Figures S12 and S13.

Reply: Thanks for the comment. We sincerely apologize for the mistakes in calculating the potentials from $E_{\text{Hg/HgO}}$ to E_{RHE} . In the last revision, we fixed the error in Figure S13. But we ignored that in Figure S12. The DEMS experiments were carried out at -0.6 V vs. Hg/HgO for the hydrogen production reaction from aldehyde and -1.3 V vs. Hg/HgO for the hydrogen evolution reaction, according to the formula:

$$E_{\text{RHE}} = E_{\text{Hg/HgO}} + 0.0591\text{pH} + 0.098$$

The E_{RHE} for the hydrogen production reaction from aldehyde should be:

$$E_{\text{RHE}} = -0.6 \text{ V} + 0.0591 \text{ V} \times 14 + 0.098 \text{ V} \approx 0.33 \text{ V}$$

And the E_{RHE} for the hydrogen evolution reaction should be:

$$E_{\text{RHE}} = -1.3 \text{ V} + 0.0591 \text{ V} \times 14 + 0.098 \text{ V} \approx -0.37 \text{ V}$$

To clarify it, Figure S12 and the caption of Figure S13 are corrected as follows:

Figure S12 Potential and current signal of HMF oxidation reaction on Au electrode in 1 M KOD and 0.5 M HMF using D_2O as the solvent at 0.33V.

Figure S13 (a) Potential and current signal and (b) Differential Electrochemical Mass Spectrometry (DEMS) signal of hydrogen evolution reaction on Au electrode in 1 M KOD and 0.5 M HMF using D_2O as the solvent at -0.37V .

2. Line 337, it should be -0.33 V instead of -0.37 V . Is the solvent D_2O ?

Reply: Thanks for the comment. According to the above calculation, the potential is -0.37 V with D_2O as the solvent. The DEMS experiments were carried out in 1 M KOD and 0.05 M HMF using D_2O as the solvent.

3. In Figure S12 and line 333, the potential should be 0.37 V instead of 0.33 V. Is the solvent D₂O?

Reply: Thanks for the comment. According to the above calculation, the potential is -0.37 V with D₂O as the solvent. The DEMS experiments are carried out in 1 M KOD and 0.05 M HMF using D₂O as the solvent.

4. Line 330, it should be Figure S12 instead of Figure S13.

Reply: Thanks for the comment. We are sorry for the error. We have changed it to “Figure S12”

5. As I remember, Baltruschat, Jusys and Heitbaum studied the hydrogen evolution mechanism during formaldehyde and acetaldehyde electrooxidation at Ib metals (Cu, Ag and Au) by using DEMS several decades ago. I suggest the authors cite them. The following is an incomplete list: *Electrochim. Acta* 2002, 47, 4485-4500 (10.1016/S0013-4686(02)00521-2); *Electrochim. Acta* 1993, 38, 1067-1072 ([https://doi.org/10.1016/0013-4686\(93\)80214-K](https://doi.org/10.1016/0013-4686(93)80214-K))

Reply: Thanks for the suggestion. Following the suggestion, we have cited these references as follows (Page 4, lines 75-76): Isotope tracing experiments showed that the hydrogen atoms of H₂ originated from aldehyde group. ^[19-24]

In the revised manuscript, the relevant references are listed as follows:

[20]H. Baltruschat, *Electrochimica Acta* 1993, 38, 1067–1072.

[21]R. Stadler, Z. Jusys, H. Baltruschat, *Electrochimica Acta* 2002, 47, 4485–4500.

[22]Z. Jusys, *J. Electroanal. Chem.* 1994, 375, 257–262.

[23]Z. Jusys, A. Vaškelis, *J. Electroanal. Chem.* 1992, 335, 93–104.

[24]M. Beltowska-Brzezinska, J. Heitbaum, *J. Electroanal. Chem. Interfacial Electrochem.* 1985, 183, 167–181.

6. In Figure 3, how were the reference spectrum/spectra taken? Please check the Y axis. Is it absorbance or reflectance? The relevant text seemed to mean that the Y axis is absorbance, however, the reflectance was shown in Figure 3. The band at 1621 cm⁻¹ could also be assigned to the HOH bending mode of water.

Reply: Thanks for the comment. The reference spectra are collected in an open circuit condition. The Y axis is absorbance. We sincerely apologize for the mistake in Figure 3. Now the Y axis of Figure 3 is corrected as “ $-\log(E_s/E_R)$ ”.

Figure 3. (a) In-situ ATR-IR spectra of HMF oxidation reactions on gold. (b) C-H scissoring of HMFMD²⁻, (c) C=C scissoring of HMFMD⁻, (d) C-H wagging of HMF, (e) C=C stretching of HMF, (f) C-H rocking of HMFCFA and (g) C=O stretching of FDCA based on the computational IR frequency analysis.

The following revision is made in the manuscript (Page 25, line 580-583): All spectra are shown in $-\log\left(\frac{E_s}{E_R}\right)$, with E_s and E_R representing the sample and reference spectra, respectively. The reference spectra are corrected in an open circuit condition, and the reference spectra are collected at different potentials.

In addition, the positive peak at 1620 cm^{-1} can indeed be attributed to the HOH bending model of water. To clarify it, the following contents have been added to the manuscript (Page 13, lines 307-309): **Additionally, a positive peak is seen at 1621 cm^{-1} between 0.43 and 0.93 V. This peak may be attributed to FDCA's C=O stretching. However, it may also be attributed to water since it is close to the bending model of water.**

Reviewer #3 (Remarks to the Author):

The authors adequately addressed all concerns. Minor comments left below:

Reply: Thanks for the valuable comment.

7. The authors discuss performance decay and how the elucidation of reaction intermediates can help solve this issue. I think this is overstated as the performance decay is often an intrinsic property of the catalyst itself rather than of specific intermediates, except in the cases of surface poisoning. Probably better to edit the arguments in the abstract/intro – I would think capturing the mechanism is more important towards modelling, designing higher performing systems and so on.

Reply: Thanks for the suggestion. Following the suggestion, we have made some revisions to the abstract and introduction parts.

In the abstract part (page 2, lines 32-33 and 43), the revision is made as follows: **However, the reaction mechanism is still not clear due to the lack of direct evidence for the critical intermediates..... unraveling a key point for designing higher performing systems.**

In the introduction part (page 4, lines 87-89), the revision is made as follows: **To promote this reaction in practical industrial applications, a thorough understanding of its reaction mechanism is essential for designing higher performing systems.**

8. From figure S4 – the amount of HMFCFA produced from non-Faradaic solution reactions is quite high (3 mM in 2 hrs). How does this quantity compare to the HMFCFA produced through Faradaic processes? This should be stated.

Reply: Thanks for the comment. The ratio of the HMFCFA produced by non-Faradaic

process and Faradaic process ranges from 7.9% (0.33V) to 0.4% (0.93V) depending on the potential. To clarify it, an additional statement is added in the manuscript (page 23, lines 529-530): **The ratio of the HMFCA generated by the non-Faradaic process to the Faradaic process ranges from 7.9% (0.33V) to 0.4% (0.93V) depending on potentials.**

9. Typos – please fix throughout text (e.g. Faradic vs Faradaic)

Reply: Thanks for the comment. We double checked the full manuscript. The typos have been fixed.

Figure 1. (a) SEM image of Au-Ni electrode. (b) XRD of Ni foam and Au-Ni electrode. (c) XPS spectra of Au-Ni electrode (d) CV curve of Au-Ni foam and Ni foam electrode in 1 M KOH with 0.5 M HMF. (e) CV curve of Au-Ni electrode in 1.0 M KOH with and without 0.5 M HMF, HMFCA, FFCA and FDCA. (f) CV curve of Au-Ni electrode in 0.01M, 0.1M and 1.0M (g) Product yield of FDCA, FFCA, DFF, HMFCA and H₂ at various potentials with the same charge. (h) Faradaic efficiency and HMFCA selectivity at various potentials of 0.33–0.93V. (i) Faradaic efficiency of two pathways at various

potentials of 0.33–0.93V. The potentials reported in this study are referred to RHE unless a special statement is made.

REVIEWERS' COMMENTS

Reviewer #2 (Remarks to the Author):

Minor revision is needed.

1. Line 324, it should be C=C stretching instead of C=C scissoring in Figure 3c.
2. Lines 582 and 583, you mentioned the reference spectra two times. I do not understand it. Please check the sentence: "The reference spectra are corrected in an open circuit condition, and the reference spectra are collected at different potentials".

Response Letter to Referees

Manuscript ID: NCOMMS-23-05443B

Dear editor and referees,

We sincerely thank all the reviewers and editor for the valuable comments and suggestions. In this Response Letter to Referees, we have addressed and clarified all the comments/concerns raised by the reviewer. With a careful revision according to the reviewer's comments, the quality of our manuscript has been improved.

Listed below are our point-by-point responses to the reviewer's comments, and all the related revision are highlighted in the revised manuscript.

REVIEWER COMMENTS

Reviewer #2 (Remarks to the Author):

Minor revision is needed.

Reply: Thanks for the careful review for our manuscript.

1. Line 324, it should be C=C stretching instead of C=C scissoring in Figure 3c.

Reply: Thanks for the comment. We are sorry for the error. We have changed it to “stretching”

2. Lines 582 and 583, you mentioned the reference spectra two times. I do not understand it. Please check the sentence: ‘The reference spectra are corrected in an open circuit condition, and the reference spectra are collected at different potentials.’

Reply: Thanks for the comment. We are sorry for the error. It should be ‘The reference spectra are corrected in an open circuit condition, and the sample spectra are collected at different potentials.’